# The Effects of High Doses of Caffeine on Maximal Strength and Muscular Endurance in Athletes Habituated to Caffeine

**DOI:** 10.3390/nu11081912

**Published:** 2019-08-15

**Authors:** Michal Wilk, Michal Krzysztofik, Aleksandra Filip, Adam Zajac, Juan Del Coso

**Affiliations:** 1Institute of Sport Sciences, Jerzy Kukuczka Academy of Physical Education, 40-065 Katowice, Poland; 2Exercise Physiology Laboratory, Camilo José Cela University, 28692 Madrid, Spain

**Keywords:** bench press, upper limb, resistance exercise, ergogenic substances, time under tension, 1RM test

## Abstract

Background: The main goal of this study was to assess the acute effects of the intake of 9 and 11 mg/kg/ body mass (b.m.) of caffeine (CAF) on maximal strength and muscle endurance in athletes habituated to caffeine. Methods: The study included 16 healthy strength-trained male athletes (age = 24.2 ± 4.2 years, body mass = 79.5 ± 8.5 kg, body mass index (BMI) = 24.5 ± 1.9, bench press 1RM = 118.3 ± 14.5 kg). All participants were habitual caffeine consumers (4.9 ± 1.1 mg/kg/b.m., 411 ± 136 mg of caffeine per day). This study had a randomized, crossover, double-blind design, where each participant performed three experimental sessions after ingesting either a placebo (PLAC) or 9 mg/kg/b.m. (CAF-9) and 11 mg/kg/b.m. (CAF-11) of caffeine. In each experimental session, participants underwent a 1RM strength test and a muscle endurance test in the bench press exercise at 50% 1RM while power output and bar velocity were measured in each test. Results: A one-way repeated measures ANOVA revealed a significant difference between PLAC, CAF-9, and CAF-11 groups in peak velocity (PV) (*p* = 0.04). Post-hoc tests showed a significant decrease for PV (*p* = 0.04) in the CAF-11 compared to the PLAC group. No other changes were found in the 1RM or muscle endurance tests with the ingestion of caffeine. Conclusion: The results of the present study indicate that high acute doses of CAF (9 and 11 mg/kg/b.m.) did not improve muscle strength nor muscle endurance in athletes habituated to this substance.

## 1. Introduction

Caffeine (CAF) is one of the most widely consumed drugs in the world and has become a popular ergogenic aid for many athletes due to its properties to improve several aspects of physical performance. The acute intake of CAF has been effective to enhance exercise performance in a wide range of sport specific tasks [1], muscular endurance [2,3,4], and strength-power exercise modalities [4,5]. The ergogenic effect of caffeine has been found when consumed at doses ranging from 3 to 9 mg/kg body mass (b.m.) and ingested in the form of capsules 30 to 90 minutes before exercise [6]. Mechanisms responsible for ergogenic effect of caffeine are linked to the impact of this substance on various tissues, organs, and systems of the human body [4,7,8,9,10]. However, there is a growing consensus to consider that caffeine’s ergogenicity lies in its tendency to bind to adenosine A_1_ and A_2A_ receptors [11]. 

Although studies have confirmed the ergogenic effects of caffeine in many aspects, much controversy remains about the effects of acute CAF intake on maximal strength (1-repetition maximum (1RM)) and local muscle endurance. Several investigations have found that the acute intake of 3–6 mg/kg/b.m. of CAF produces an increase in 1RM test performance [3,12,13,14], and in the total number of repetitions performed (T-REP) [12,13,15]. However, other investigations have found that the same dosage did not produce such effects [2,3,5,15,16], suggesting that other factors such as the type of testing, the muscle mass involved, and the athlete’s experience in strength training might affect the ergogenic effect of caffeine on muscle performance. Furthermore, Wilk et al. [2] observed a positive effect of CAF intake on time under tension (TUT) in a muscle endurance test, but no significant effect in the T-REP. According to Wilk et al. [17] and Burd et al. [18], TUT might be the most reliable indicator to assess exercise volume in resistance exercise regardless of the number of repetitions performed. Based on literature review, it can be concluded that previous results of studies on the acute effects of CAF intake on muscle strength and endurance are inconclusive. 

Most investigations on the effects of caffeine intake on muscle performance have used participants unhabituated to caffeine or with low-to-moderate daily consumption of caffeine from 58 to 250 mg/day [3,12,16]. However, caffeine is an ergogenic aid frequently used in training and competition and it seems that athletes seeking for caffeine ergogenicity are already habituated to caffeine. There are reports indicating that 75–90% of athletes consume CAF before or during training sessions and competitive events [19,20,21], which indicates that studies on the effectiveness of acute CAF intake are particularly important in habitual caffeine users.

According to Svenningsson et al. [22] and Fredholm et al. [23], habitual caffeine intake modifies physiological responses to acute ingestion of CAF by the up-regulation of adenosine receptors. Furthermore, constant exposure to caffeine could impact caffeine metabolism by inducing an accelerated conversion of caffeine into dimethylxanthines by the cytochrome P450. Therefore, progressive habituation to the performance benefits of caffeine intake has been recognized in humans when it is consumed chronically [24]. However, the evidence to certify the existence of habituation to the ergogenic benefits is still inconclusive because it was found that low caffeine consumers benefited from the acute intake of 3–6 mg/kg/b.m. of CAF to a similar extent as individuals habituated to caffeine [25,26]. Lara et al. [27] found that caffeine ergogenicity was lessened when the substance was ingested daily (3 mg/day/kg/b.m.) for 20 days but it was still ergogenic after this period. In contrast, Beaumont et al. [28] observed that caffeine’s ergogenicity practically disappeared after 28 days of daily ingestion (1.5–3 mg/day/kg/b.m.). Interestingly, all these investigations tested tolerance to caffeine’s ergogenicity using endurance exercise protocols, while only one of them used muscle performance tests. Wilk et al. [5] showed that neither 3, 6, nor 9 mg/kg/b.m. of CAF intake enhanced power output and bar velocity during bench press exercise in strength-trained male athletes habituated to caffeine. However, there are no available data regarding the influence of acute CAF intake on maximal strength and muscular endurance in athletes habitually consuming caffeine. 

Due to the aforementioned contrasting results, the main goal of this study was to assess the acute effect of high doses of CAF (9 and 11 mg/kg/b.m.) on maximal strength and muscle endurance assessed on the basis of T-REP and TUT in athletes habituated to CAF (4–6 mg/day/kg/b.m.). We hypothesized that high doses of caffeine, exceeding athletes’ usual daily consumption of caffeine, would enhance muscle strength and muscular endurance. Since the value of daily habitual intake of caffeine may significantly modify the acute ergogenic effects of CAF ingestion, we used doses of CAF significantly above daily consumption in this investigation.

## 2. Materials and Methods

### 2.1. Study Participants

Sixteen healthy strength-trained male athletes (age: 24.2 ± 4.2 years, body mass: 79.5 ± 8.5 kg, body mass index (BMI): 24.5 ± 1.9, bench press 1RM: 118.3 ± 14.5 kg; mean ± standard deviation) volunteered to participate in the study after completing an ethical consent form. Participants had a minimum of 3 years of strength training experience (4.1 ± 1.4 years) and practiced team sports. All participants were classified as high habitual caffeine consumers according to the classification recently proposed by Gonçalves et al. [26]. The participants self-reported their daily ingestion of CAF (4.9 ± 1.1 mg/kg/b.m., 411 ± 136 mg of caffeine per day) based on the Food Frequency Questionnaire (FFQ) with their average consumption assessed for four weeks before the start of the experiment. The inclusion criteria were as follows: (a) Free from neuromuscular and musculoskeletal disorders, (b) performance of the bench press exercise with a load of at least 120% of body mass, (c) habitual caffeine intake in the range of 4–6 mg/day/kg/b.m., ~300–500 mg of caffeine per day. Participants were excluded when they suffered from any pathology or injury. Additionally, they were required to refrain from alcohol and tobacco consumption and were asked not to take any medications or dietary supplements as well as other ergogenic substances during and two weeks prior to the experiment. The study protocol was approved by the Bioethics Committee for Scientific Research at the Academy of Physical Education in Katowice, Poland, according to the ethical standards of the latest version of the Declaration of Helsinki, 2013.

### 2.2. Habitual Caffeine Intake Assessment

Habitual caffeine intake was assessed by an adapted version of the Food Frequency Questionnaire (FFQ) proposed by Bühler et al. [29]. The FFQ was completed individually with the supervision of a qualified nutritionist. The FFQ was employed to assess the habitual consumption of dietary products containing caffeine. Portions, in household measures, were used to assess the amount of food consumed according to the following frequency of consumption: a) More than three times a day, b) two to three times a day, c) once a day, d) five to six times a week, e) two to four times per week, f) once a week, g) three times per month, h) rarely or never. The list was composed of dietary products with moderate-to-high caffeine content including different types of coffee, tea, energy drinks, cocoa products, popular beverages, medications, and caffeine supplements. Previously published information and nutritional tables were used for database construction [1,30,31]. Based on the answers in the FFQ, a qualified nutritionist estimated the habitual caffeine intake for each participant.

### 2.3. Experimental Design

This study used a randomized, double-blind, placebo-controlled crossover design where each participant acted as his own control. Participants performed a familiarization session with a preliminary 1RM test on one day and three different experimental sessions with a one-week interval between sessions to allow complete recovery and substances wash-out. The blinding and randomization of the sessions was conducted by a member of the research team that was not directly involved in data collection. 

During the three experimental sessions, participants either ingested a placebo (PLAC), 9 mg/kg/b.m. of CAF (CAF-9) or 11 mg/kg/b.m. of CAF (CAF-11). After 60 minute of absorbing the substances, participants underwent a 1RM strength test and a muscle endurance test with the bench press exercise. During each test, power output and bar velocity were measured. Both CAF and PLAC were administered orally 60 minute before each exercise protocol to allow peak blood caffeine concentration and at least 2 hours after the last meal to maintain the same time of absorption. CAF was provided in the form of capsules containing the individual dose of CAF (Caffeine Kick®, Olimp Laboratories, Dębica, Poland). The producer also prepared identical PLAC capsules filled out with an inert substance (all-purpose flour). Participants refrained from physical activity the day before testing and they kept their habitual training routines during the study period. In addition, participants were instructed to maintain their usual hydration and dietary habits during the study period including habitual caffeine intake and register their calorie intake using “Myfitness pal” software [32] every 24 hours before the testing procedure. The average calorie intake was ~3300 kcal/day and it was similar before the three experimental trials. Participants were also asked to refrain from caffeine intake 12 hours before each trial. All testing was performed at the Strength and Power Laboratory at the Jerzy Kukuczka Academy of Physical Education in Katowice, Poland.

### 2.4. Familiarization Session and One Repetition Maximum Test

A familiarization session preceded the preliminary one repetition maximum testing. Participants arrived at the laboratory at the same time of day as in the upcoming experimental sessions (in the morning, between 9:00 and 10:00). Upon arrival, participants cycled on an ergometer for 5 minutes at an intensity that resulted in a heart rate of approximately 130 bpm, followed by a general upper body warm-up. Next, participants performed 15, 10, 5, and 3 repetitions of the bench press exercise using 20, 40, 60, and 80% of their estimated 1RM with a 2/0/X/0 tempo of movement. The sequence of digits describing the tempo of movement (2/0/X/0) referred to a 2 seconds eccentric phase, 0 represented a pause during the transition phase, X referred to the maximum possible tempo of movement during the concentric phase, and the last digit indicated no pause at the end of movement [33]. Participants then executed single repetitions of the bench press exercise with a 5 minutes rest interval between successful trials. The load for each subsequent attempt was increased by 2.5 to 5 kg, and the process was repeated until failure. Hand placement on the barbell was individually selected with a grip width on the barbell of 150% individual bi-acromial distance (BAD). BAD was determined by palpating and marking the acromion with a marker, and then measuring the distance between these points with a standard anthropometric tape. The positioning of the hands was recorded to ensure consistent hand placement during all testing sessions. No bench press suits, weightlifting belts, or other supportive garments were permitted. Three spotters were present during all attempts to ensure safety and technical proficiency. 

### 2.5. Experimental Protocol

Three testing sessions were used for the experimental trials and the protocols were identical. All testing took place between 9.00 and 11.00 to avoid circadian variation. The general warm-up for the experimental sessions was identical to the one used for the familiarization session. After warming up, participants performed the 1RM bench press test to assess upper-body maximal muscle strength. For the 1RM test, the first warm-up set included eight to ten repetitions with 50% 1RM determined during the familiarization session. The second set included three to five repetitions with 75% 1RM. Participants then completed one repetition with 95% 1RM. Based on whether the participant successfully lifted the load or not, the weight was increased or decreased (2.5 to 5 kg) in subsequent attempts until the 1RM value for the session was obtained. Three- to five–minute rest intervals were allowed between the 1RM attempts, and all 1RM values were obtained within five attempts. After a five-minute rest interval, muscle endurance was assessed with one ‘all-out’ set using a load of 50% of participants’ 1RM measured in the previous 1RM test. The end of the muscle endurance test was assumed when momentary concentric failure occurred. The concentric phase of the bench press movement was performed at maximal possible velocity in each repetition, while the eccentric phase was performed with 2 seconds duration (2/0/X/0). During the muscle endurance test, the following variables were registered: T-REP—total number of repetitions [n];TUT_CON_—time under tension of concentric contractions [s];PP—peak concentric power [W];MP—mean concentric power [W];PV—peak concentric velocity [m/s];MV—mean concentric velocity [m/s].

All repetitions were performed without bouncing the barbell off the chest, without intentionally pausing at the transition between the eccentric and concentric phases, and without raising the lower back off the bench. During the experimental trials, participants were encouraged to perform at maximal effort according to the recommendations by Brown and Weir [34]. A linear position transducer system (Tendo Power Analyzer, Tendo Sport Machines, Trencin, Slovakia) was used for the evaluation of bar velocity. The Tendo Power Analyzer is a reliable system for measuring movement velocity and to estimate power output [35,36]. The system consists of a velocity sensor connected to the load by a Kevlar cable which, through an interface, instantly transmits the vertical velocity of the bar to specific software installed in the computer (Tendo Power Analyzer Software 5.0). The system measures upward vertical mean and peak velocity of the movement. Using a set external load, the system calculates mean and peak power output in the concentric phase of the movement. The measurement was made independently in each repetition and automatically converted into the values of power (max, mean) and concentric velocity (max, mean). All familiarization and experimental sessions were recorded by means of a Sony camera (Sony FDR191 AX53). Time under tension and the number of performed repetitions was obtained manually from the recorded data using slow speed playback (1/5 speed). In order to ensure the reliability of manual data collection, four independent observers performed data analysis from the Sony camera. There were no significant differences in TUT [s] nor in T-REP [n] between the data collected by 4 evaluators. All participants completed the described testing protocol that was carefully replicated in the subsequent experimental sessions. 

### 2.6. Side Effects

Immediately after finishing testing, and after 24 hours, participants answered a side effects questionnaire (QUEST), which is a nine-item measure with a dichotomous (yes/no) response scale of caffeine ingestion [20,37,38].

### 2.7. Statistical Analysis

The Shapiro-Wilk, Levene, and Mauchly´s tests were used in order to verify the normality, homogeneity and sphericity of the sample data variance. Verification of differences between the PLAC vs. CAF-9 and CAF-11 groups was performed using one-way ANOVA. In the event of a significant main effect, post-hoc comparisons were conducted using the Tukey’s test. Percent relative effects and the 95% confidence intervals were also calculated. Effect Sizes (Cohen’s *d*) were reported where appropriate. Parametric effect sizes (ES) were defined as large for *d* > 0.8, moderate between 0.8 and 0.5, and small for <0.5 [39]. Statistical significance was set at *p* < 0.05. All statistical analyses were performed using Statistica 9.1 and were presented as means ± standard deviations.

## 3. Results

The one-way ANOVA revealed a statistically significant difference in PV (*p* = 0.04; Table 1) between PLAC vs. CAF-9 and CAF-11 groups. However, no significant differences in 1RM, T-REP, TUT_CON_, MP, PP, nor MV between PLAC, CAF-9, and CAF-11 groups were observed among experimental sessions. Next, the Tukey’s post-hoc test revealed a significantly lower PV in the CAF-11 when compared to the PLAC group (*p* = 0.04; Table 2). 

### Side Effects

Table 3 details the occurrence of nine different side effects assessed immediately after and 24 hours after testing. Immediately after the PLAC trial, participants reported a very low frequency of side effects (0–13%; QUEST + 0 hour). After CAF-9 ingestion, there were more severe side effects (0–88%; QUEST + 0 hour) compared to the PLAC trial. The most severe side effects were recorded for increased urine output, tachycardia and heart palpitations, anxiety or nervousness, perception of performance improvement, and increased vigor (63–88%; QUEST + 0 hour). Finally, the CAF-11 trial produced a drastic increase in the intensity and frequency of side effects (0–92%; QUEST + 0 hour; Table 3). 

In the morning following testing (QUEST + 24 hours), very few participants (0–13%) reported side effects with the PLAC. The CAF-9 trial showed greater frequency of side effects (0–69%), with increased urine output, tachycardia and heart palpitations, gastrointestinal problems, and increased vigor in comparison with the PLAC trial. Finally, CAF-11 intake increased the frequency and severity of all adverse side effects, with a frequency of appearance from 0 to 88% (Table 3).

## 4. Discussion

The main finding of the study was that, compared to the ingestion of the PLAC, the acute intake of high doses of CAF (9 and 11 mg/kg/b.m.) was not effective to produce any statistically measurable ergogenic effect on the bench press 1RM, T-REP, TUT_CON_, PP, MP, nor MV in individuals habituated to CAF intake. In fact, the intake of 11 mg/kg/b.m. significantly decreased PV during bench press testing performed to concentric muscle failure in these habitual caffeine users. All this information suggests that even high doses of CAF were ineffective to produce ergogenic effects on maximal strength and muscular endurance in high-caffeine consumers. This lack of effect was evident despite the fact that the dosage of caffeine used pre-exercise was well-above their daily intake of this substance. In addition, these data might be indicative of tolerance to caffeine’s ergogenicity for muscle performance while the high occurrence of side effects is still maintained with high doses of caffeine.

Previous studies have shown a variety of effects when different doses of CAF were administered to athletes performing testing to assess maximum strength and muscle endurance. Some of them indicated a significant increase in 1RM and T-REP performance [12,13], while others did not confirm such benefits [2,14]. Perhaps differences in the results of previous studies may be attributed to different doses of CAF consumed by study participants, in addition to the use of participants with an uneven habituation to caffeine. Since the value of daily habitual intake of caffeine might significantly modify the acute ergogenic effects of CAF ingestion [40], this investigation was aimed to study the acute effects of high doses (9 and 11 mg/kg/b.m.) of CAF intake on maximal strength and muscle endurance of the upper limbs, using athletes clearly habituated to caffeine. 

Previous research using well-controlled caffeine treatments has suggested that the habitual intake of this stimulant might progressively reduce the ergogenic effect of acute CAF supplementation on exercise performance [27,28,40], reductions after acute CAF intake in habitual users can be modified using pre-trial doses which should be greater than the daily habitual intake. However, our results do not support this statement. Despite the fact that the doses of CAF used in our study were much greater (9 and 11 mg/kg/b.m.) than the daily intake of studied athletes (4–6 mg/kg/b.m./day), there were no positive changes in the analyzed strength, endurance, and power variables. In fact, our results indicate a significant decrease in PV after the intake of CAF-11 compared to the PLAC. Previous studies showed that acute CAF intake leads to higher activation of motor units [41] and higher MVIC [10,42]. However in the presented study the supposed effect of increased muscle tension following CAF intake, not only did not increase the power output generated during the CON phase of the movement, but a decrease in PV was observed. A decrease in PV after ingestion of CAF-11 undermines the legitimacy of using high doses of CAF before explosive, high-velocity, low-resistance exercises performed to muscle failure. According to Pallarés et al. [37], explosive, high-velocity, low-resistance actions require a much lower CAF dose (3 mg/kg/b.m.) in individuals with none or low habituation to caffeine. However, in the light of the current results, this statement does not apply to habitual caffeine users. The results of the present study, and especially the decrease in PV after CAF intake (11 mg/kg/b.m.), are particularly important for competitive athletes, since research indicates that 75–90% of athletes consume CAF before or during training sessions and competitive events [19,20]. In this regard, when seeking the benefits of acute caffeine intake to muscle performance, the dishabituation to caffeine should be recommended instead of the use of doses above the daily intake of caffeine. For how long habitual caffeine users should discontinue the intake of caffeine merits further investigation. For now, current evidence suggests that the dishabituation period should be longer than four days [43].

Furthermore, besides statistically significant change in PV, the results of the study showed negative effect sizes (ES) and relative (%) decreases in T-REP, TUT_CON_, MP, PP, PV, and MV after the intake of CAF-11 compared to the PLAC, as well as relative decreases in MP, PP, PV, MV following the ingestion of CAF-9 compared to the PLAC. Decreased values of T-REP and TUT_CON_ after acute intake of CAF-11 may have resulted from the increased muscle tension generated during the movement [10,42]. A supposed increase of muscle activation can lead to a higher energy demand during exercise, thus leading to a faster depletion of energy substrates in muscle cells [44], which may partially explain a decline in T-REP and TUT_CON_ after the intake of CAF-11. However, the increased muscle tension following CAF intake did not improve the power output generated during the CON phase of the movement. The relative increase in results was observed only in the 1RM test after the intake of CAF-9 and CAF-11 (3.3% and 4.7%, respectively) and in TUT_CON_ after consuming CAF-9 (10.5%). While such an improvement in results of the 1RM test may be considered small in statistical terms, it can be of great significance in training and competition of elite athletes, especially in competitions where success depends on maximal strength production [45]. The relative % increase in results of the 1RM test after the intake of CAF-9 and CAF-11 compared to the PLAC is partly compatible with Pallarés et al. [37] who demonstrated that muscle contractions against heavy loads (75–90% 1RM) required a high CAF dose (9 mg/kg/b.m.) to obtain an ergogenic effect in low caffeine consumers. The results of our research confirm that, also in habitual consumers, high doses of CAF ingestion might be effective in improving maximal strength, although this effect is accompanied by a high occurrence of side effects (Table 3). Additionally, the TUT_CON_ increased by 10.5% after the intake of CAF-9 compared to the PLAC what may be of great significance in training of elite strength athletes. However, the increase in TUT_CON_ in the present study is contrary to the results of Wilk et al. [2], who showed a decrease in TUT during the bench press exercise at 70% 1RM performed to muscle failure after the intake of CAF (5 mg/kg/b.m.) compared to the PLAC. It should be pointed out that differences in the external load used in both exercise protocols (50% 1RM vs. 70% 1RM) could have affected the results following CAF intake [37]. The 10.5% increase in TUT_CON_ in the present study indicates that TUT may be an additional indicator of training volume during resistance training, compared to the T-REP, where a 0.4% decrease in results was registered after the intake of CAF-9 compared to the PLAC. 

Furthermore, the results of our study showed that high doses of CAF in habitual caffeine consumers may be ineffective or also have a negative effect on physical performance in athletes. The ingestion of CAF-9 and CAF-11 significantly increased the frequency of self-reported side effects (0–88% for CAF-9; 0–92% for CAF-11) compared to the PLAC. It has been empirically established that side effects of caffeine intake are severe when doses between 9 and 13 mg/kg/b.m. are used [46]. Increased urine output, tachycardia and heart palpitations, anxiety or nervousness, as well as perception of performance improvement are among the most common adverse effects experienced by athletes when they consume caffeine [47]. The current investigation adds some valuable information as it indicates that these adverse effects are still persistent in individuals habituated to caffeine, at least when they consume a high dose of CAF to exceed their habitual intake of this substance. However, the occurrence of these side effects does not always prevent athletes from improving their performance, as was the case with rowers in Carr et al. [48], who improved their times in a 2000-m ergometer test, or participants in the study of Pallarés et al. [37], who significantly improved their neuromuscular performance after the ingestion of 9 mg/kg/b.m. of CAF. On the contrary, Wilk et al. [5] showed an increased frequency of all adverse side effects after the intake of 9 mg/kg/b.m. of CAF yet with no significant increases in power output and bar velocity during the bench press exercise compared to the PLAC. All this information might be indicative of the necessity of evaluating both performance and side effects when planning to use >9 mg/kg/b.m. of CAF before training or competition. 

The duration of adverse effects resulting from CAF intake is another issue to be considered in research and sports training. The present study showed a drastic increase in the reported frequency of side effects 24 hours after ingestion (Table 3). The CAF-9 trial showed a frequency of side effects in the range from 0 to 69%, with increased urine output, tachycardia and heart palpitations, gastrointestinal problems, as well as increased vigor in comparison with the PLAC group. CAF-11 intake increased the frequency of all adverse side effects, with a frequency of appearance from 0 to 88%. It should be stressed that even if caffeine allows for improved physical performance, it can significantly disturb sleep indices at night, such as sleep efficiency and ability to fall asleep, as well as induce an overall decrease in sleep itself [49]. Therefore, athletes who consume CAF to enhance their performance during training and/or competition should take into account its detrimental effects on sleep, especially if subsequent high-intensity exercise is to be performed on the following day. 

The present study has several limitations which should be addressed. The procedure of the research assumed all participants were similarly habituated to caffeine despite the fact that their daily intake of caffeine and the duration of this intake presented some inter-individual variation. It has been recently suggested that all individuals respond to caffeine ingestion when caffeine is compared to a placebo using multiple and repeated testing sessions [50]. Although two different tests were used to assess the effect of caffeine intake on muscle performance, the ergogenic effect of CAF was not evident, suggesting that habituation to caffeine precluded the effect of acute CAF intake. However, it is still possible that the use of other muscle strength tests can still show ergogenic effects of high doses of CAF on performance of caffeine-habituated athletes. Furthermore, there were no genetic assessments related to caffeine metabolism in the tested athletes. According to Cornelis et al. [51], genetic variation in the A_2A_ receptor (ADORA2A), the main target of caffeine action in the central nervous system, is associated with caffeine consumption. The probability of having the ADORA2A 1083TT genotype associated with caffeine-induced anxiety decreases as the caffeine intake increases in a population, and subjects with that genotype are more likely to limit their caffeine intake. People who were homozygous for the 1083T allele experienced greater anxiety after consuming 150 mg of caffeine [52]. Before the start of our experiment, no study participant reported any side effects after consumption of CAF within the last six months suggesting that the side effects found in this investigation were the result of the high doses used in this study rather than a genetic predisposition.

### Practical Applications

The ingesting of high doses of CAF (9 and 11 mg/kg/b.m.) can bring minor benefits during training with near or maximal external loads. However, if explosive, high-velocity, low-resistance exercises are performed to muscle failure, the high doses of CAF (9 and 11 mg/kg/b.m.) are not recommended as they may hinder performance. These suggestions apply only to habitual strength-trained male caffeine users. 

## 5. Conclusions

The results of the present study indicate that acute intake of high doses of CAF (9 and 11 mg/kg/b.m.) before exercise did not produce significant improvements in maximal strength and muscle endurance during the bench press exercise performed to concentric failure in a group of habitual caffeine users. However, it should be noted that slight benefits in 1RM and TUT_CON_ after the intake of high doses of CAF were observed. In addition, the results of this study showed a significant decrease in PV of the bar after the intake of CAF-11 compared to the PLAC. Overall, this investigation indicates that the use of high doses of CAF does not improve significant performance during resistance exercises in high caffeine consumers while it causes a significant increase in the occurrence of side effects. These outcomes undermine the convenience of using high doses of CAF before resistance training performed to momentary muscle failure. However, these results may not translate to other forms, volumes, or intensities of exercise. Future research should compare the inter-subject variation in response to different doses of caffeine. Additionally, as Chtourou and Souissi [53] mention, it would be wise to compare the changes in power-output and strength responses to CAF intake between several time-points following ingestion.

## Figures and Tables

**Table 1 nutrients-11-01912-t001:** Summary of performance data under the three employed conditions.

Variable	Placebo (95% CI)	CAF-9 (95% CI)	CAF-11 (95% CI)	F	*p*
1RM [kg]	118.3 ± 14.5 (109.4–125.5)	122.3 ± 15.3 (115.7–132.5)	124.2 ± 11.4 (116.3–135.2)	0.24	0.78
T-REP [n]	25.1 ± 3.2 (23.3–26.8)	25.0 ± 4.9 (22.4–27.6)	25.6 ± 3.3 (23.8–27.3)	0.09	0.90
TUT_CON_ [s]	17.1 ± 3.29 (15.3–18.8)	19.1 ± 3.29 (17.3–20.8)	16.9 ± 3.39 (15.1–18.8)	2.01	0.14
MP [W]	348 ± 79 (305–390)	333 ± 72 (294–372)	318 ± 78 (276–360)	0.61	0.54
PP [W]	798 ± 164 (710–886)	766 ± 134 (694–837)	731 ± 186 (632–831)	0.61	0.51
MV [m/s]	0.71 ± 0.10 (0.66–0.76)	0.67 ± 0.08 (0.63–0.72)	0.70 ± 0.07 (0.66–0.74)	0.8	0.45
PV [m/s]	1.39 ± 0.16 (1.31–1.48)	1.37 ± 0.15 (1.29–1.45)	1.25 ± 0.17 (1.16–1.34)	3.43	0.04 *

All data are presented as mean ± standard deviation; CI—confidence interval; * statistically significant difference *p* < 0.05; 1RM: One repetition maximum; T-REP: Total number of repetitions; TUT_CON_: Time under tension during concentric movement; MP: Mean power output; PP: Peak power output; MV: Mean velocity; PV: Peak velocity.

**Table 2 nutrients-11-01912-t002:** Differences in placebo vs. caffeine conditions between experimental trials.

Variable	Comparison	*p*	Effect Size (Cohen *d*)	Relative Effects [%]
1RM [kg]	Placebo vs CAF-9	0.82	0.26—small	3.3 ± 4.1
	Placebo vs CAF-11	0.74	0.45—small	4.7 ± 5.1
T-REP [n]	Placebo vs CAF-9	0.99	−0.02—negative effects	0.4 ± 12.1
	Placebo vs CAF-11	0.93	0.15—small	2.0 ± 11.2
TUT_CON_ [s]	Placebo vs CAF-9	0.22	0.6—moderate	10.5 ± 15.5
	Placebo vs CAF-11	0.99	−0.05—negative effects	−6.2 ± 21.5
MP [W]	Placebo vs CAF-9	0.85	−0.19—negative effects	−1.5 ± 7.6
	Placebo vs CAF-11	0.51	−0.38—negative effects	−9.4 ± 10.5
PP [W]	Placebo vs CAF-9	0.84	−0.21—negative effects	−4.2 ± 8.3
	Placebo vs CAF-11	0.48	−0.38—negative effects	−9.2 ± 11.6
MV [m/s]	Placebo vs CAF-9	0.43	−0.44—negative effects	−6.0 ± 11.8
	Placebo vs CAF-11	0.91	−0.11—negative effects	−1.4 ± 6.6
PV [m/s]	Placebo vs CAF-9	0.90	−0.12—negative effects	−1.5 ± 10.2
	Placebo vs CAF-11	0.04 *	−0.84—negative effects	−11.2 ± 10.7

All data are presented as mean ± standard deviation; * statistically significant difference *p* < 0.05; 1RM: One repetition maximum; T-REP: Total number of repetitions; TUT_CON_: Time under tension during concentric movement; MP: Mean power output; PP: Peak power output; MV: Mean velocity; PV: Peak velocity.

**Table 3 nutrients-11-01912-t003:** Number (frequency) of participants that reported side effects immediately after the testing protocol (side effects questionnaire (QUEST) + 0 hour) and 24 hours later (QUEST + 24 hours).

Side Effects	Occurrence of Side Effects in Particular Groups					
	PLAC		CAF-9		CAF-11	
	+0 h	+24 h	+0 h	+24 h	+0 h	+24 h
Muscle soreness	0 (0%)	0 (0%)	0 (0%)	0 (0%)	0 (0%)	0 (0%)
Increased urine output	1 (6%)	1 (6%)	10 (63%)	9 (57%)	10 (63%)	10 (63%)
Tachycardia and heart palpitations	3 (19%)	1 (6%)	12 (76%)	11 (69%)	15 (92%)	13 (81%)
Anxiety or nervousness	1 (6%)	2 (13%)	11 (69%)	4 (25%)	14 (88%)	13 (81%)
Headache	2 (13%)	1 (6%)	3(19%)	6 (37%)	8(50%)	8 (50%)
Gastrointestinal problems	0 (0%)	1 (6%)	6 (38%)	10(63%)	6 (38%)	13 (81%)
Perception of performance improvement	2 (13%)	0 (0%)	14 (88%)	0 (0%)	6 (38%)	0 (0%)
Increased vigor/activeness	2 (13%)	1 (6%)	13(81%)	8 (50%)	6 (38%)	6 (38%)
Insomnia	0 (0%)	0 (0%)	0 (0%)	4 (25%)	0 (0%)	6 (38%)

Data are presented as the number of participants (frequency) that responded affirmatively to the existence of a side effect.

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
