# Peer review of "The Effects of High Doses of Caffeine on Maximal Strength and Muscular Endurance in Athletes Habituated to Caffeine"

_nutrients, 2019, doi:10.3390/nu11081912_

Round 1

Reviewer 1 Report

Many thanks for the opportunity to read this manuscript. It is an interesting study, however, my major concern is the very high dose of caffeine and the relation with the side effects. 

I also think that the inter-subject variation in the response to caffeine should be presented as part of the background and discussed further. 

A couple of questions/comments?

Was there any washout period between trials? Line 19: b.m. needs to be defined before using the abbreviation. Line 27: I would recommend presenting some data here Line 77: there is a "/" that needs to be removed Line 97: Was it self-reported? Line 132: Did you measure blood caffeine concentration? Were the sleep habits/patterns measured or recorded before the trials? In tables 1 and 2 define the abbreviations in the legend.  I think in the discussion, more attention could be given to some mechanistic detail, I felt it lacked in-depth.  I would also recommend to include some of the practical implications of the study?

Author Response

Dear Reviewer

We are very grateful for your constructive comments and suggestions that helped us improve our manuscript. Please find below changes that we have done to the manuscript according to your suggestions.

Many thanks for the opportunity to read this manuscript. It is an interesting study, however, my major concern is the very high dose of caffeine and the relation with the side effects. 

Point 1 - I also think that the inter-subject variation in the response to caffeine should be presented as part of the background and discussed further. 

Reply - We agree with the reviewer that the inter-subject variation in the response to caffeine may show significant differences compared to group analysis, that's why we will perform the inter-subject analysis in a separate article. However we added a sentence regarding this matter in the limitations of the study.

The added sentence L 397-400:

Future research should compare the inter-subject variation in response to different doses of caffeine. Additionally  as Chtourou and Souissi mention (2012), it would be wise to compare the changes in power-output and strength responses to CAF intake between several time- points following ingestion.

A couple of questions/comments?

Point 2 -Was there any washout period between trials?

Reply – the participants were also asked to refrain from caffeine intake 12 hours before each trial.

Point 3 -Line 19: b.m. needs to be defined before using the abbreviation.

Reply - Changes have been made

Point 4 -line 27: I would recommend presenting some data here

Reply - Changes have been made

Point 5 -Line 77: there is a "/" that needs to be removed

Reply - Changes have been made

Point 6 -Line 97: Was it self-reported?

Reply - self-reported refers to the average consumption of  caffeine by the study participants assessed four weeks before the start of the experiment. Habitual caffeine intake was assessed by an adapted version of the Food Frequency Questionnaire (FFQ).

Version before changes:

All participants were classified as high habitual caffeine consumers as per the classification recently proposed by Gonçalves et al. [26] and they reported daily ingestion of CAF (4.9 ± 1.1 mg/kg/b.m., 411 ± 136 mg of caffeine per day) based on their average consumption assessed for four weeks before the start of the experiment.

Changed to (L 97-101):

All participants were classified as high habitual caffeine consumers according to the classification recently proposed by Gonçalves et al. [26]. The participants self-reported daily ingestion of CAF (4.9 ± 1.1 mg/kg/b.m., 411 ± 136 mg of caffeine per day) was based on the Food Frequency Questionnaire (FFQ) and their average consumption assessed for four weeks before the start of the experiment.

Point 7 -Line 132: Did you measure blood caffeine concentration? Were the sleep habits/patterns measured or recorded before the trials?

Reply - No, blood CAF concentration and sleep habits/patterns were not measured.

Point 8 -In tables 1 and 2 define the abbreviations in the legend. 

Reply – Changes have been made

Point 9 -I think in the discussion, more attention could be given to some mechanistic detail, I felt it lacked in-depth. 

We agree with the reviewer and we added a new sentences

New sentences (L 286-289)

Previous studies showed that acute CAF intake leads to higher activation of motor units (Duncan et al., 2014) and higher MVIC (Behrens et al., 2015, Park et al., 2008). However in the presented study the supposed effect of increased muscle tension following CAF intake, not only did not increase the power output generated during the CON phase of the movement, but a decrease in PV was observed.  

Second new sentences (L 305-311)

Decreased values of T-REP and TUTCON after acute intake of CAF-11 may have resulted from the increased muscle tension generated during the movement (Behrens et al., 2015, Park et al., 2008). A supposed increased of muscle activation can lead to a higher energy demand during exercise, thus leading to a faster depletion of energy substrates in muscle cells (Bogdanis 2012), which may partially explain a decline in  T-REP and TUTCON after the intake of CAF-11. However, the increased muscle tension following CAF intake did not improve the power output generated during the CON phase of the movement.

Point 10 -I would also recommend to include some of the practical implications of the study?

We agree with the reviewer and Practical Applications was added

New sentences  (L 380-385)

Practical Applications

The ingesting of high doses of CAF (9 and 11 mg/kg/b.m.) can bring minor benefits during  training with near or maximal external loads. However if explosive, high-velocity, low-resistance exercises are performed to muscle failure the high doses of CAF (9 and 11 mg/kg/b.m.) are not recommended as they may hinder performance. These suggestions apply only to habitual strength-trained male caffeine users.

Reviewer 2 Report

General comments

The present study compares the intake of 9 and 11 mg/kg/b.m of CAF on maximal strength and muscle endurance in healthy strength-trained male athletes habituated to caffeine. Using a randomized, double-blind, placebo-controlled crossover design, each participant performs three test sessions, i.e., PLAC, CAF-9 and CAF-11, with one week in-between. For healthy strength-trained male athletes, the authors concluded that CAF-9 and CAF-11 did not improve muscle strength nor muscle endurance in habitual caffeine consumers’.

The manuscript is well written and present interesting findings. However, some modifications and/or explanations are needed:

Introduction

L.88. For the hypothesis, I suggest Adding an explanation of the rational of the utilization of 11 mg/kg/b.m (utilized or not in previous study, etc.).

Participants

Did the author perform a-priori calculation of the sample size? If yes, please add this information.

Statistical analysis

I suggest adding the confidence interval.

Discussion/limitations

As a previous review study (Chtourou and Souissi, 2012 – JSCR) confirmed that muscle strength and power are higher in the late afternoon, I suggest adding that future studies could perform this protocol in the afternoon or compare the responses between two or more time-of-day points.

Specific comments

L.28. remove “statistically”.

L.34. Utilize capital letters for the first letter of the keywords.

Author Response

Dear Reviewer 1,

We are very grateful for your constructive comments and suggestions that helped us improve our manuscript. Please find below changes that we have done to the manuscript according to your suggestions.

General comments

The present study compares the intake of 9 and 11 mg/kg/b.m of CAF on maximal strength and muscle endurance in healthy strength-trained male athletes habituated to caffeine. Using a randomized, double-blind, placebo-controlled crossover design, each participant performs three test sessions, i.e., PLAC, CAF-9 and CAF-11, with one week in-between. For healthy strength-trained male athletes, the authors concluded that CAF-9 and CAF-11 did not improve muscle strength nor muscle endurance in habitual caffeine consumers’.

The manuscript is well written and presents interesting findings. However, some modifications and/or explanations are needed:

Introduction

Point 1 - L.88. For the hypothesis, I suggest adding an explanation of the rational of the utilization of 11 mg/kg/b.m (utilized or not in previous studies, etc.).

Changes have been made

Version before changes:

We hypothesized that high doses of caffeine, exceeding athletes’ usual daily consumption of caffeine, would enhance muscle strength and muscular endurance.

Changed to (L 87-91):

We hypothesized that high doses of caffeine, exceeding athletes’ usual daily consumption of caffeine, would enhance muscle strength and muscular endurance. Since the value of daily habitual intake of caffeine may significantly modify the acute ergogenic effects of CAF ingestion, therefore in this investigation we used doses of CAF, significantly above daily consumption.

Participants

Point 2 -Did the author perform a-priori calculation of the sample size? If yes, please add this information.

Reply – the a-priori calculation of the sample size wasn’t performed.

Statistical analysis

Point 3 -I suggest adding the confidence interval.

Information about confidence interval has been included in the tables.

Discussion/limitations

Point 4 -As a previous review study (Chtourou and Souissi, 2012 – JSCR) confirmed that muscle strength and power are higher in the late afternoon, I suggest adding that future studies could perform this protocol in the afternoon or compare the responses between two or more time-of-day points.

Reply - As suggested, a new sentence has been added (L 397-400)

Future research should compare the inter-subject variation in response to different doses of caffeine. Additionally  as Chtourou and Souissi mention (2012), it would be wise to compare the changes in power-output and strength responses to CAF intake between several time- points following ingestion.

Specific comments

Point 5 -L.28. remove “statistically”.

Reply – Changes have been made

Point 6 -L.34. Utilize capital letters for the first letter of the keywords.

Reply – Changes have been made